# Foliar Aspersion of Salicylic Acid Improves Phenolic and Flavonoid Compounds, and Also the Fruit Yield in Cucumber (*Cucumis sativus* L.)

**DOI:** 10.3390/plants8020044

**Published:** 2019-02-16

**Authors:** Pablo Preciado-Rangel, Juan José Reyes-Pérez, Silvia Citlaly Ramírez-Rodríguez, Lilia Salas-Pérez, Manuel Fortis-Hernández, Bernardo Murillo-Amador, Enrique Troyo-Diéguez

**Affiliations:** 1Torreón Technological Institute (National Technologic of México—ITT), Torreón, Coahuila 27170, Mexico; ppreciador@yahoo.com.mx (P.P.-R.); fortismanuel@hotmail.com (M.F.-H.); 2Technical State University of Quevedo, Av. Quito km 1.5 vía a Santo Domingo de los Tsáchila, Quevedo, Los Ríos, Ecuador 120501, Ecuador; jreyes@uteq.edu.ec; 3Postgraduate Program, Torreón Technological Institute (TNM—ITT), Torreón, Coahuila 27170, Mexico; citlaly_rrha@hotmail.com; 4Facultad de Contaduría y Administración, Universidad Autónoma de Coahuila, Fco. Javier Mina 150, Luis Echeverría Álvarez Sector Norte, Torreón, Coahuila 27085, Mexico; lilia-net@hotmail.com; 5Center for Biological Research of Northwest México (CIBNOR S.C.), La Paz 23096, B.C.S. Mexico; bmurillo04@cibnor.mx

**Keywords:** elicitor, nutraceutical quality, flavonoid, *Cucumis sativus* L., salicylic acid

## Abstract

The aim of this research is to evaluate the effect of foliar application of salicylic acid (SA) on the yield and phytochemical content in hydroponically grown cucumber (*Cucumis sativus* L.). (1) Background: The importance of Mexico’s cucumber production is based on its cultivation in recent decades as one of the major winter crops; in addition, the production of vegetables under hydroponic systems has increased significantly during the last few years, with cucumber being one of the vegetables with a high economic potential. (2) Methods: A completely randomized experimental design with 15 repetitions was used. SA at five doses (0.075, 0.1, 0.15, 0.25, and 0.5 mM) and one control (distilled water) was sprinkled weekly on cucumber plants. The evaluated variables were yield (total fruit weight per plant), fruit parameters (length, size and firmness), and nutraceutical quality of cucumber. (3) Results: Low concentrations of SA improve the yield and high concentrations decrease it, but the nutraceutical quality of fruits is improved, as compared to the control treatment. (4) Conclusions: In order to obtain a higher content of bioactive compounds without affecting the yield and commercial quality of cucumber fruits, it is advisable to use the average concentration (0.15 mM) of SA.

## 1. Introduction

In Mexico, the production of vegetables under hydroponic systems in greenhouses has increased significantly during the last few years, being cucumber (*Cucumis sativus* L.) one of the vegetables with a high economic potential, because of its high consumption index, both, as fresh and industrialized fruit, occupying 10% of the total area of greenhouses in the country [1]. These production systems have allowed for increases in yield, which is not necessarily accompanied by a high nutraceutical quality of the harvested products [2]. Nutraceutical quality is a term that is attributed to a high content of bioactive compounds in a food that have the characteristic of preventing chronic degenerative diseases by inhibiting the oxidation of cells, contributing to cell maintenance and antioxidant balance [3]. These compounds have been used as food ingredients, giving rise to so-called functional foods, which provide a health benefit superior to that of traditional nutrients [4].

Cucumber fruits are a good source of mineral compounds and dietary fiber, contain approximately 95% water, 3.6% carbohydrates, and 0.65% protein, and are low in calories [5]. They are a good source of vitamin C (8 mg/100 g), retinol equivalent (28 μg/100 g), vitamin E (0.16 mg/100 g) and B-group vitamins, including folic acid (15 μg/100 g), pantothenic acid, and phenolic and flavonoid compounds [6].

In recent years, much attention has been given to the control of greenhouse management practices to improve nutraceutical quality of horticultural crops, including irrigation and nutrition, which are cultural practices that can further affect the flavonoid content in vegetables [7], with elicitation being a widely used agronomic technique. This practice allows increasing or inducing the biosynthesis of phytochemicals in plants as a result of the induction of signaling pathways helping to avoid the decrease in yield and promoting the formation of bioactive compounds [8]. Elicitation can be promoted by a variety of compounds that are externally applied, allowing the mentioned physiological and phytochemical benefits. A widely used compound is salicylic acid (SA), called by some authors signaling molecule [9], since it has been shown to improve performance [10], prolonging the useful life in fruits [11] and increasing the biosynthesis of phytochemicals [12].

Therefore, the objective of this research work was to determine the effect of foliar application of SA on the performance and nutraceutical quality of cucumber fruits developed under hydroponic conditions.

## 2. Results

### 2.1. Yield

Foliar sprays of SA in concentrations of up to 0.015 mM in cucumber plants increased fruit yield up to 39.10%, while higher concentrations decreased it (Table 1). This increase in yield is closely related to the increase in the number of fruits per plant [13], increase in the foliar area [14], and in the root system [15], which improves the absorption of nutrients and finally the yield. Shakirova et al. (2003) [16] mentioned that the positive effect of SA on growth and yield may be due to its influence on other plant hormones, since SA alters the balance of auxins, cytokinins and abscisic acid, increasing growth and performance. In contrast, high doses decrease the yield of crops, because they cause an oxidative stress that the plant is unable to restore [17,18].

### 2.2. Commercial Quality and Firmness of Fruit

The quality of cucumber fruits depends on the firmness and size, once the harvested fruits are undergoing changes at the morphological and physiological levels, especially in the metabolism, which influences the appearance and integral quality of the product that reaches the final consumer [19].

A greater firmness of fruits confers a longer storage life and the ability to resist shocks during transport, storage or marketing [20]. Leaf spraying of SA affected the firmness of the fruits (Table 1). The concentrations of 0.075 to 0.1 mM increased the firmness of the fruits in relation to the control fruits. Shafiee et al. (2010) [21] mentioned that the SA decreases the softening of the fruits, since it causes a lower production of ethylene, inhibiting the action of enzymes responsible for the degradation of the cell wall, such as polygalacturonase (PG), cellulase (CEL) and pectin-metillesterase (PME) which are ethylene-dependent [22]. A fruit with little firmness is susceptible to rapid wilting [23] and dehydration, as well as the formation of a spongy tissue and less turgor, due to the loss of water from the cells by perspiration, a product of plasmolysis and the lower accumulation of sugars in cell walls [24].

### 2.3. Fruit Size

The size of the fruits was significantly affected by the different doses of SA (Table 1), the dose of 0.075 mM of SA exceeded by 13.345% and 7.27% in length and diameter the fruits obtained by the control, respectively (Figure 1). A previous study reported that SA acts as a growth regulator that accelerates cell division and can increase the size, number of fruits and yield [25]. In contrast, high doses of SA decrease fruit size probably due to an inhibitory effect of cell division caused by a hormonal imbalance [26].

### 2.4. Nutraceutical Quality

The phenolic compounds and flavonoids are an immense group of secondary metabolites of plants that are found naturally and that have a wide variety of advantages when consumed in human food, giving food nutraceutical quality [3]. In this work, the nutraceutical quality of the fruits was affected by the SA foliar spray (Table 1).

## 3. Discussion

With the increase in the concentration of SA, the concentration of total phenolic compounds and total flavonoids was improved, obtaining the highest concentration of these compounds at a dose of 0.5 mM, exceeding the control treatment by 37.5% and 45.46%, respectively. El-Gaied et al. (2013) observed that phytochemical compounds are increased in response to a biochemical stress in cell suspensions caused by the increase in SA [27]. It has also been found that SA induces the production of hydrogen peroxide (H_2_O_2_), which stimulates a greater activity of phenylalanine ammonium lyase, responsible for the synthesis of phenolic compounds [28]. Mora-Herrera et al. (2011) suggested that the SA in the plants activates the secondary metabolism and increases the synthesis of phytochemical compounds, as well as the synthesis of antioxidants [29]. Accordingly, Sánchez-Chávez et al. (2011) reported an increase of phenolic and total flavonoids in chili fruits with application of SA [30], whereas Vázquez-Díaz et al. (2016) obtained a higher concentration of phenolic and flavonoids in tomato fruits due to the increase in SA doses [31]. On the other hand, soybean plants with different SA treatments increase the pigment content, as well as the level of photosynthates [32,33].

In addition, SA exerts an influence on antioxidant activity and a decrease in lipid peroxidation, which gives nutraceutical quality to the fruit. In this sense, it has been reported that SA has an effect on the photosynthetic capacity, this being attributed to the stimulation of the RuBisCO and to the accumulation of pigments [33], resulting in a higher content of pigments in plants treated with SA, which coincides with the results obtained in the present study where it was confirmed that there was a higher phenolic and flavonoid content in fruits in which SA was applied. The concentrations of these compounds will vary according to the SA applied, the application technique used, as well as the evaluated plant species [34]. However, the tendency of its increase in response to the application of SA is an agronomic tool that can be used to increase the nutraceutical quality of the fruits; in addition, the use of SA is a relevant topic, since it can increase the content of antioxidants, and presumably may improve the metabolism of carbohydrates, although this assumption should be further studied. On the other hand, research on the precise pathway of SA biosynthesis, mode of action and other key points leading to the synthesis of phenolic and flavonoid compounds [35] is still being developed.

## 4. Materials and Methods

### 4.1. Plant Material and Growth Conditions

The study was performed in an automatic tunnel type greenhouse, covered with plastic, with a humid module with extractors, covering an area of 144 m^2^, located in the Torreón Technological Institute in Coahuila, Northern Mexico. Fresh type cucumber cultivar Poinset (Nunhems^®^) was produced under greenhouse conditions, and this variety is the most used by producers under greenhouse environment. Cucumber seedlings were transplanted when seedlings were 15–20 cm high and had 2 real leaves, and were placed in 20 L black plastic bags as pots, which contained river sand and vermiculite (80:20) for ensuring a sandy texture and adequate porosity of the hydroponic substrate; one seedling per pot was established. River sand was previously washed and sanitized using a 5% sodium hypochloride solution; then, pots were sorted in a single line separated by 0.4 m between plants and by 1.4 m between rows, for a plant density of 35,714 plants ha^−1^.

Pollination was performed manually every day at 12:00–14:00, from flowering start to fruit development. Plants were pruned to one single stem attached with string to the greenhouse ceiling, while fruits were placed in plastic mesh pockets tied to the support structure. A drip irrigation system was used to irrigate plants and provide nutrition with the nutrient solution Steiner (1984) [36]. Plants were sprayed with 0.5 L pot^–1^ day^–1^ three times, from transplant to flowering, and with 1.0 L pot^−1^ day^−1^, from flowering to harvest; such volumes were previously verified to ensure enough wetting. During the growth period, all axillary vegetative buds were removed from the main stem, and periodically the stem was attached to the twine with plastic plant clips.

### 4.2. Experimental Design and Treatments

Treatments were established in a completely randomized experimental design using 15 plants per treatment, with each plant being a treatment replicate. Salicylic acid (SA-2 hydroxybenzoic acid) obtained from Sigma Chemical Co. (St. Louis, U.S.A.) was initially dissolved in 100 mL of dimethyl sulfoxide (pH 6.0–6.5) prepared with distilled water containing 0.02% Tween 20 (Polyoxyethylene sorbitan) [13]. The SA treatments consisted in 0.075, 0.1, 0.15, 0.25, and 0.5 mM solutions, and a control (distillated water). These SA solutions were applied to the plants every 8 days by spraying during the whole cultivation period, taking place between 8:00 and 10:00 am using a hand-held sprayer. In order to avoid interferences with different moisture levels, the same amount of distilled water was sprayed to the control plants at a given time. The lower leaf surface was sprayed until wetted as the upper surface since it was reported that absorption by the lower leaf surface was rapid and effective [37]. The purpose to carry out applications by means of several sprays instead of a single dose was to ensure absorption; in this sense, the efficacy of foliar sprays is largely dependent on three prime factors, soil, plant and environmental factors, of which those related to the plant are quite relevant, but further assessments should be carried out about timely applications.

### 4.3. Analytical Assessments

The evaluated variables were yield (total product weight per plant), produce quality parameters (length, diameter and firmness), and nutraceutical quality of cucumber. Quality parameters were determined using 10 fruits per treatment.

### 4.4. Extracts Preparation for Nutraceutical Quality

The nutraceutical quality of the fruits was measured as the total content of phenolic and flavonoid compounds. Five grams of fresh pulp were mixed in 10 mL of ethanol in a plastic tube with screw cap, which was placed on a rotatory shaker (ATR Inc., EU) for 6 h at 5 °C and 20 rpm. Afterwards, the tubes were centrifuged at 3000 rpm for 5 min, and the supernatant was extracted for analytical tests [38].

### 4.5. Content of Total Phenolic Compounds

The total phenolic content was measured using the Folin–Ciocalteau method [39]. Accordingly, 300 μL of sample were mixed and added with 1080 mL of distilled water and 120 μL of Folin–Ciocalteau reagent (Sigma-Aldrich, St. Louis MO, USA), vortexing for 10 s. After 10 min, 0.9 mL of sodium carbonate (7.5% w/v) was added, stirring for 10 s. The solution was allowed to stand at room temperature for 30 min, and then its absorbance at 765 nm was read in an HACH 4000 spectrophotometer. The phenolic content was calculated by means of a standard curve using gallic acid (Sigma, St. Louis, Missouri, USA) as a standard; results were reported in mg of equivalent gallic acid per 100 g of fresh weight (mg equiv GA·100 g^−1^ FW), which is the main indicator for this variable. The analyses were performed in triplicate.

### 4.6. Total Flavonoid Content

The total flavonoid content was determined using a colorimetric method [40]. For this purpose, 250 μL of the ethanolic extract was mixed with 1.25 mL distillated water in a test tube followed by addition of 75 μL of 5% NaNO_2_ solution. After 5 min, 150 μL of 10% AlCl_3_ + H_2_O solution were added and allowed to stand for another 6 min; then, a volume of 500 μL of 1M NaOH was added, plus additional 275 μL distilled water. All components were mixed by vortexing. The absorbance was measured immediately at 510 nm using a spectrophotometer (Genesys 10UV). The results were expressed in mg equivalents of quercetin per 100 g^−1^ based on fresh weight (mg equiv Q·100 g^−1^ FW).

### 4.7. Length, Diameter and Fruit Firmness

The length and diameter were measured with a lab caliper; weight was measured with a precision balance; firmness was measured using an Extech penetrometer (FHT200) with a 7 mm plunger (Figure 2), according to the thickness of fresh pulp. For this purpose, the fruits were placed on a hard and fixed surface, registering the average of two measurements per fruit, for each repetition and treatment.

### 4.8. Statistical Analysis

Data of response variables were analyzed by an analysis of variance, and mean comparisons were conducted with the Tukey test (*p* ≤ 0.05).

## 5. Conclusions

Foliar spray of SA affects the yield, firmness, size, length and nutraceutical quality of cucumber fruits produced under hydroponic greenhouse conditions. Doses of 0.075 mM SA increase the yield and commercial quality of fruits, but not the nutraceutical quality. Doses of SA at 0.5 mM decrease the yield, while commercial quality decreases; however, the content of bioactive compounds increases considerably, thus increasing the nutraceutical quality of the fruits. In other applications, it was found that a pre-treatment with 0.5 mM SA can alleviate chilling injury symptoms effectively and maintain fruit quality in cucumber [41]. According to our results, although the maximum and minimum observed values were 94.8 and 57.8 mg·100 g^−1^ FW, respectively, with the doses of 0.50 and 0.075 mM of SA, while the control treatment (0.0 mM) yielded 51.7 mg·100 g^−1^ FW, and the use of SA in low concentrations is a viable alternative to increase of performance and commercial quality. In order to obtain a higher content of bioactive compounds without affecting the yield and commercial quality of cucumber fruits, it is advisable to use medium doses (0.15 mM) of SA.

## Figures and Tables

**Figure 1 plants-08-00044-f001:**
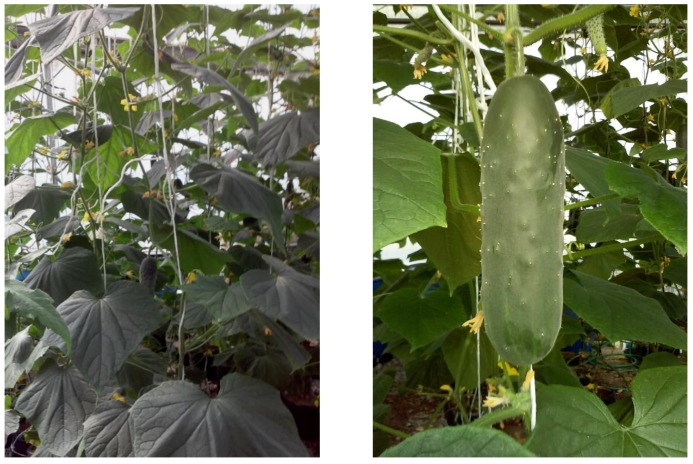
Cucumber plants and fruit, harvested in a tunnel type greenhouse, in Torreón, Coahuila State, Mexico.

**Figure 2 plants-08-00044-f002:**
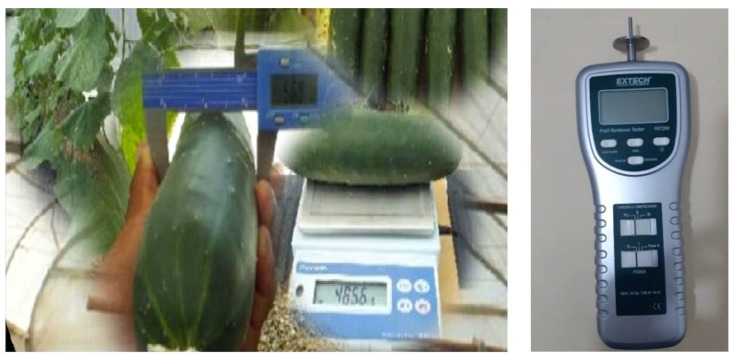
Lab caliper and precision weighing balance (left); Extech penetrometer (FHT200) (right).

**Table 1 plants-08-00044-t001:** Average values for variables evaluated in cucumber fruits, with the application of SA in different concentrations.

Treatment	Yield	Firmness	Length	Diameter	Total Phenolics	Total Flavonoids
(mM)	(kg)	(N)	(cm)	(cm)	(mg·100 g^–1^ FW)	(mg·100 g^–1^ FW)
0.00	2.18 ± 0.17b *	16.4 ± 1.2b *	19.3 ± 1.4b *	5.1 ± 0.10b *	105.0 ± 3.4d *	51.7 ± 3.9d *
0.075	3.58 ± 0.22a	33.6 ± 2.2a	22.3 ± 1.8a	5.5 ± 0.13a	107.7 ± 5.5d	57.8 ± 4.1cd
0.10	3.54 ± 0.16a	33.3 ± 4.5a	22.2 ± 2.3a	5.3 ± 0.28ab	119.8 ± 6.3c	62.2 ± 3.3c
0.15	3.54 ± 0.17a	33.0 ± 2.2a	19.9 ± 0.5ab	5.1 ± 019b	129.8 ± 6.4b	77.2 ± 4.5b
0.25	1.57 ± 0.31c	19.1 ± 2.9b	19.3 ± 0.7b	4.7 ± 0.18c	134.8 ± 7.1b	77.7 ± 5.7b
0.50	1.41 ± 0.31c	17.3 ± 2.5b	19.2 ± 0.4b	4.6 ± 0.16c	168.7 ± 4.0a	94.8 ± 6.4a

* Values with equal letters within each column are statistically similar (*p* ≤ 0.05).

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
