# Peer review of "Foliar Aspersion of Salicylic Acid Improves Phenolic and Flavonoid Compounds, and Also the Fruit Yield in Cucumber (Cucumis sativus L.)"

_plants, 2019, doi:10.3390/plants8020044_

Round 1

Reviewer 1 Report

Salicylic acid is represented by authors as SA in some parts of the manuscript (lines 65, 69, 70) while in other sections they used AS (lines 92, 93, 94, 95). They should use SA in the whole manuscript. This mistake is all over the manuscript

In line 18 "content" is redundant

At the end of line 20, size or font seems to be changed

Line 51and 52 are not very well connect. Writting should be improved

In lines 243 and 244 they refer to results of other authors work. I Think that they can use that information in discussion section, but in conclusion they should focus on the finding related directly with their data

I missed the method to measure firmness

Authors do not explain why they used several application of salicylic acid instead of just a single dose. What was the criterion?

Line 142 I think that authors cannot support with their data that SA improved carbohydrates metabolism, even antioxidant capacity because they did not measure it. I would have been a good idea that they had measured it

Line 124 H2O2 should be H2O2

Author Response

RESPONSES TO REVIEWER 1

    1)  Salicylic acid is represented by authors as SA in some parts of the 

         manuscript (lines 65, 69, 70) while in other sections they used AS (lines

         92, 93, 94, 95). They should use SA in the  whole manuscript. This 

         mistake is all over the manuscript

Response: It was corrected and made it uniform (SA)

2)       In line 18 "content" is redundant

Response: It was deleted and redundancy eliminated

3)       At the end of line 20, size or font seems to be changed

Response: It was verified and corrected

4)       Line 51and 52 are not very well connect. Writting should be improved

Response: It was re-expressed

5)       In lines 243 and 244 they refer to results of other authors work. I Think that they can use that information in discussion section, but in conclusion they should focus on the finding related directly with their data

Response: It was arranged and simplified

6)       I missed the method to measure firmness

Response: It was included

Length and diameter were measured with a lab caliper; weight with a precision balance; firmness was measured using an Extech penetrometer (FHT200) with a 7 mm plunger (Figure 2). For this purpose, the fruits were placed on a hard and fixed surface, registering the average of two measurements per fruit, for each repetition and treatment.

7)       Authors do not explain why they used several application of salicylic acid instead of just a single dose. What was the criterion?

Response:  explained and justified

8)       Line 142 I think that authors cannot support with their data that SA improved carbohydrates metabolism, even antioxidant capacity because they did not measure it. I would have been a good idea that they had measured it

Response:  It was explained

...However, the tendency of its increase in response to the application of SA is an agronomic tool that can be used to increase the nutraceutical quality of the fruits; in addition, the use of SA is a relevant topic, since it can increase the content of antioxidants, and presumably may improve the metabolism of carbohydrates, although this assumption should be further studied. 

9)       Line 124 H2O2 should be H2O2

Response: It was corrected

Thanks for suggestions.

Reviewer 2 Report

The manuscript provides interesting information on the effect of the application of salicylic acid on cucumber. However, the authors talk about an increase in the quality of the fruit, but they do not contribute data referring to visual quality, for example, color, sensory evaluation ... It would be interesting for the authors to include this information

The method to measure firmness is not in the manuscript

Author Response

RESPONSES TO REVIEWER 2

1)       The manuscript provides interesting information on the effect of the application of salicylic acid on cucumber. However, the authors talk about an increase in the quality of the fruit, but they do not contribute data referring to visual quality, for example, color, sensory evaluation ... It would be interesting for the authors to include this information

Response: Of course, these parameters are quite important; we will consider your kind suggestions in next studies

2)       The method to measure firmness is not in the manuscript

Response: It was included

.....firmness was measured using an Extech penetrometer (FHT200) with a 7 mm plunger (Figure 2). For this purpose, the fruits were placed on a hard and fixed surface, registering the average of two measurements per fruit, for each repetition and treatment.

Reviewer 3 Report

Interesting research. I suggest some minor comments to improve the paper:

Please clarify the volume of SA applied (May be drop point?, as reported by Mejia-Teniente et al. 2019, Physiological and Molecular Plant Pathology usually used with several elicitors). This is to describe in a wider manner the application of elicitors for readers.

Please check some typo mistakes throughout the text,I.e. AS is not correct the correct is SA ( check it please).

Figure 2 is not necessary.

Author Response

RESPONSES TO REVIEWER 3

1)     Please clarify the volume of SA applied (May be drop point?, as reported by Mejia-Teniente et al. 2019, Physiological and Molecular Plant Pathology usually used with several elicitors). This is to describe in a wider manner the application of elicitors for readers.

Response: It was re-expressed and explained.

Plants were sprayed 0.5 L pot-1 day-1 three times per day, from transplant to flowering, and 1.0 L pot-1 day-1, from flowering to harvest; such volumes were previously verified to ensure enough wetting.

2)     Please check some typo mistakes throughout the text,I.e. AS is not correct the correct is SA ( check it please).

Response: They were corrected and made them uniform

3)     Figure 2 is not necessary.

Response: It was deleted; instead, a set of pertinent photographs were included
